# Genomic Methylated Cytosine Level during the Dedifferentiation and Cellular Competence in *Coffea arabica* Lines: Insights about the Different In Vitro Responses

João Paulo de Morais Oliveira [1,*], Natália Arruda Sanglard [1], Adésio Ferreira [2] and Wellington Ronildo Clarindo [1,3]

1 Laboratório de Citogenética e Cultura de Tecidos Vegetais, Centro de Ciências Agrárias e Engenharias, Universidade Federal do Espírito Santo, Alegre 29500-000, ES, Brazil; nataliasanglard@gmail.com (N.A.S.); welbiologo@gmail.com (W.R.C.)

2 Laboratório de Biometria, Departamento de Agronomia, Universidade Federal do Espírito Santo, Alegre 29500-000, ES, Brazil; adesioferreira@gmail.com

3 Laboratório de Citogenética e Citometria, Departamento de Biologia Geral, Centro de Ciências Biológicas e da Saúde, Universidade Federal de Viçosa, Viçosa 36570-900, MG, Brazil

* Correspondence: joaopaulo.ueg@gmail.com

**Abstract:** *Coffea arabica* genotypes present distinct responses in vitro, and somaclonal variation occurrence has been reported. Global cytosine methylation is one of the epigenetic mechanisms that influences the *Coffea* in vitro responses. We aimed to establish the indirect somatic embryogenesis in *C. arabica* 'Catuaí Vermelho', 'Caturra' and 'Oeiras', associate the distinct responses to the methylated cytosine genomic level, and check the ploidy stability. Leaf explants were cultured in callus induction and proliferation medium. The resulted calli were transferred to the regeneration medium, and the mature cotyledonary somatic embryos were transferred to the seedling medium. 'Oeiras' exhibited the highest number of responsive leaf explants, followed by 'Caturra' and 'Catuaí Vermelho'. Global methylated cytosine level increased over time in the 'Catuaí Vermelho' and 'Caturra' friable calli, remaining constant in 'Oeiras'. 'Oeiras' did not regenerate somatic embryos, while 'Catuaí Vermelho' exhibited the highest number. Somatic embryo regeneration was associated with the increase of the methylated cytosine level. However, the 'Catuaí Vermelho' embryogenic calli showed a lower methylated cytosine level than 'Caturra'. Recovered plantlets exhibited the same 2C value and chromosome number to the explant donors. Therefore, cytosine hypermethylation occurred during *C. arabica* indirect somatic embryogenesis, influencing cell competence and somatic embryos regeneration.

**Keywords:** coffee; epigenetic; indirect somatic embryogenesis; plant tissue culture; ploidy level

## 1. Introduction

Coffee breeding programs have sought to develop and select more productive genotypes that are resistant to biotic and abiotic stress and adapted to the soil conditions of marginal areas and with less environmental impact [1]. As the coffee plants are perennial and the conventional breeding requires 30 to 35 years to select superior genotypes, corresponding to six or seven cycles of self-pollination, the use and improvement of other strategies is fundamental [1]. In this context, plant biotechnology coupled with conventional breeding has contributed to genetic breeding in an attempt to create and develop superior genotypes [2,3]. The plant tissue culture applied to coffee breeding is an important biotechnological tool since, through in vitro culture, it is possible to: (a) propagate superior genotypes on a large scale [4]; (b) maintain, conserve and exchange germplasm [5]; (c) obtain homozygous plantlets from another culture [6]; (d) induce polyploidization [7] and somaclonal variations [8]; (e) regenerate transgenic cells or tissues [9]; and (f) overcome incompatibility barriers through protoplast fusion [10,11].

Indirect somatic embryogenesis (ISE) is an in vitro morphogenetic pathway that consists of cultivating segments of plant tissue in a sterile and specific culture medium, where callus is formed and, subsequently, somatic embryos (SE) giving rise to plantlets [12–14]. The capacity of the plant cell to organize and give rise to a new plant identical to the explant donor plant is based on the concept of cellular totipotency, postulated by Haberlandt in 1902 [15]. Cellular totipotency involves complex mechanisms of gene expression reprograming, cellular division and metabolism, and cellular development reprogramming [16]. The genetic [5], epigenetic [17,18] and physiological and/or morphological [15,19] aspects of the explant donors, as well as the conditions in vitro [4,20], influence ISE.

In addition to the characteristics inherent to explant donors and the in vitro environment, the epigenome also influences ISE [21–23]. Epigenetic variations are defined as covalent modifications that occur in chromatin, allowing cells to maintain distinct and different characteristics, although they contain the same genetic material [14,21,24,25]. Variations of the global methylation levels of genomic DNA have been reported in plant cells grown in vitro during the processes of cell dedifferentiation and differentiation in *Coffea canephora* and 'Híbrido de Timor' [22,23,26]. In *C. canephora*, cytosine hypermethylation has been associated with SE conversion and maturation via direct [22] and indirect embryogenesis [23]. The same conclusion was reported by [26] from natural allotriploid and synthetic autoallohexaploid 'Híbrido de Timor' (*C. canephora* × *Coffea arabica*). Regarding this data for *Coffea*, the evaluation of the genomic methylated cytosine in *C. arabica* may explain the intraspecific variations reported for this species during in vitro establishment.

In vitro propagation of *Coffea* via ISE has been associated with genetic (nuclear or organelar) and/or epigenetic instability, which is induced by tissue culture environment [8,20,27]. The phenomenon of genomic and epigenomic changes are denominated as somaclonal variation [28,29]. These variations (loss of genetic fidelity) are associated with genomic (nucleus, mitochondria and plastid) and/or epigenomic changes, which can result in phenotypic variations in the regenerated plantlets in relation to the explant donor plant [26,29]. Somaclonal variation has been identified and evaluated from chromosome number changes [7,26,30], chromosomal rearrangements [31], mutations in DNA sequences [19,20,30,32] and/or histone modifications [33].

In this study, we used three lines of *C. arabica*, 'Catuaí Vermelho', 'Caturra' and 'Oeiras', which originated from genetic recombination, mass selection and pedigree (breeding methods applied for segregating populations obtained from the crossing between two lines), respectively, to understand the genotypic influence on the morphogenic response in vitro and to understand the role of global cytosine methylation. Thus, the present study aimed to: (a) establish and compare the indirect somatic embryogenesis in *C. arabica* 'Catuaí Vermelho', 'Caturra' and 'Oeiras' under the same in vitro condition, (b) verify for influence of the global genomic level of methylated cytosine on the responses during dedifferentiation and cellular competence, and (c) check the stability of the regenerated plantlets in vitro by counting the chromosome number and content of nuclear DNA 2C.

## 2. Materials and Methods

### 2.1. ISE Establishment

Completely expanded leaves of *C. arabica* 'Caturra', 'Oeiras' and 'Catuaí Vermelho' were collected from orthotropic nodes of plants that have been maintained in the germplasm bank of the Universidade Federal de Viçosa (UFV), Minas Gerais, Brazil, 20°45' S, 42°52' W, under adequate phytosanitary and environmental conditions. The collected leaves were used as explant source for ISE.

The collected leaves were washed with liquid detergent and rinsed with running water for 2 h, and then disinfected in a laminar flow chamber by immersion in 70% ethanol for 20 s, and 1.5% sodium hypochlorite solution for 20 min [34]. Posteriorly, leave explants of 1–2 cm$^2$ were excised, and five fragments were cultured in Petri dishes containing callus induction medium (M1, Supplement 1). The callus formation was evaluated biweekly until 90 days, and after, the calli were individually transferred to SE regeneration medium (M2,

Supplement 1). The Petri dishes were kept in the dark at $25 \pm 2$ °C. Mature cotyledon somatic embryos (MCSE) regeneration was evaluated monthly up to 180 days. The MCSE were counted and transferred to the seedling recovery medium (M3, Supplement 1). In each test tube, one MCSE was inoculated. Subsequently, the test tubes were maintained in a growth room under a light/dark regime of 16/8 h with 36 µmol m$^{-2}$ s$^{-1}$ of light radiation supplied by two fluorescent lamps (20 W, Osram®) at $25 \pm 2$ °C.

### 2.2. Genomic Methylated Cytosine Level

Friable callus samples from 'Catuaí Vermelho', 'Caturra' and 'Oeiras' were separately collected after 60 and 90 days in callus induction and proliferation medium. Embryogenic callus samples (callus showing an embryogenic response) from 'Catuaí Vermelho' and 'Caturra' were also separately collected after MCSE formation in SE regeneration medium. Then, the collected samples were macerated separately in the MagNALyser (Roche®, Germany) for 60 s at 7000 rpm. Genomic DNA was extracted according to [35] with the addition of 7.5 M ammonium acetate and excluding the nocturnal period for DNA precipitation [7]. DNA concentration, quality and purity were determined using the NanoDrop spectrophotometer (Thermo Scientific® 2000c) and DNA integrity was assessed by 0.8% agarose gel electrophoresis.

For the measurement of the 5-methylcytosine, 30 µg of genomic DNA were diluted in 100 µL of dH$_2$O. Then, 50 µL of 70% (*v/v*) perchloric acid were added to the diluted DNA, and then hydrolyzed for 1 h in a 100 °C water bath. The pH of the hydrolysates was adjusted between 3–5 with KOH (1 M). Cytosine and 5-methylcytosine standard stock solutions were prepared by weighing 0.44 mg cytosine and 0.55 g 5-methylcytosine and dissolving it in 0.1% perchloric acid. Posteriorly, concentrations of cytosine and 5-methylcytosine stock solution were $4 \times 102$ µmol L$^{-1}$ and 20 µmol L$^{-1}$, respectively. 5-methylcytosine was analyzed according to [26,36] and by High Performance Liquid Chromatography (Shimadzu® HPLC, model LC-20AT), which is equipped with a photodiode array detector (SPD–M20A) using a silica-based reverse phase column C18 (VP–150 × 4.6 mm, 5 µm). The sample separations were carried out under a specific mobile phase of potassium dihydrogen phosphate (50 mmol L$^{-1}$, pH 5.8) at a flow rate of 0.5 mL min$^{-1}$, in which each component of the sample interacts in a different way with the sorbent material, generating different speeds and leading to separation as they run through the column. Cytosine and its methylated derivate detection were performed at a wavelength of 285 nm. The genomic methylated cytosine level was calculated using the equation: 5-methylcytosine level = [5-methylcytosine/(cytosine + 5-methylcytosine)] × 10, comparing the values with cytosine and 5-methylcytosine standards.

### 2.3. Ploidy Level Stability

Leaf fragments (2 cm$^2$) from in vitro regenerated plantlets of *C. arabica* 'Catuaí Vermelho' and 'Caturra' and of the internal standard *Solanum lycopersicum* L. (2C = 2.00 pg, 38) were co-chopped in nuclei extraction buffer [37,38]. The suspensions were processed, stained [5] and analyzed on a flow cytometer (Partec® GmbH, Muenster, Germany). The 2C value was measured considering the G0/G1 nuclei peak of the 'Catuaí Vermelho' or 'Caturra' and of *S. lycopersicum*. In addition, the root meristem of the regenerated plantlets of 'Catuaí Vermelho' and 'Caturra' were collected, washed, fixed and enzymatically macerated. The slides were prepared using cell dissociation and air-drying techniques [5]. After staining with 5% Giemsa, mitotic images were captured using a 100× objective and a CCD camera (Nikon EvolutionTM) connected to a Nikon 80i microscope (Nikon, Japan). From the mean 2C value and chromosome number, the DNA ploidy level was confirmed for the regenerated 'Catuaí Vermelho' and 'Caturra' plantlets.

### 2.4. Statistical Analysis

ISE responses of *C. arabica* 'Catuaí Vermelho', 'Caturra' and 'Oeiras' were compared during callus induction/proliferation and during SE regeneration stages. In the callus

induction and proliferation stage, statistical analysis was performed using the mean number of responsive explants, which were defined by the presence of calli at 15, 30, 45, 60, 75 and 90 days, totaling 45 repetitions for 'Caturra', 60 for 'Oeiras' and 102 for 'Catuaí Vermelho'. To compare the mean number of responsive explants of 'Catuaí Vermelho', 'Caturra' and 'Oeiras' over time, the data were transformed $\sqrt{}$ (x + 0.5), analysis of variance (ANOVA) was performed and the mean values compared by Tukey's test ($p < 0.05$). Quadratic polynomial regression analysis ($p < 0.05$) was performed, due to it being better adjusted to the observed mean values. During the SE regeneration stage, an ANOVA test was also performed for 'Catuaí Vermelho' and 'Caturra', using the mean number of MCSE regenerated at 30, 60, 90, 120, 150 and 180 days, totaling 116 repetitions for 'Caturra' and for 'Catuaí Vermelho'. The data were transformed $\sqrt{}$ (x + 0.5), ANOVA was performed, and the mean values were compared by the *F* test ($p < 0.05$) and displayed with box-plot graphs. Then, a regression analysis was performed at 5% of the probability level ($p < 0.05$). Quadratic polynomial regression analysis ($p < 0.05$) was performed, due to it being better adjusted to the observed mean values.

The genomic methylated cytosine level was measured during the callus induction/proliferation and the SE regeneration stages. ANOVA was applied to compare the genomic methylated cytosine level of the 'Catuaí Vermelho', 'Caturra' and 'Oeiras' friable calli collected at 60 (8 repetitions for 'Catuaí Vermelho', 9 for 'Caturra' and 20 for 'Oeiras') and 90 days (6 repetitions for 'Catuaí Vermelho' and 'Caturra' and 20 for 'Oeiras') in callus induction and proliferation medium. Mean values were compared by Tukey's test ($p < 0.05$). To compare the genomic methylated cytosine level in 'Catuaí Vermelho' (7 repetitions) and 'Caturra' (3 repetitions) friable calli in SE regeneration medium, ANOVA was applied, and mean values were compared by the *F* test ($p < 0.05$). All analyzes were performed using the software R.

## 3. Results

### 3.1. ISE Establishment

*C. arabica* 'Catuaí Vermelho', 'Caturra' and 'Oeiras' presented distinct mean values of responsive explants (callus in the leaf explant), statistically differing over time (Figure 1A). The first responsive explants were observed at 15 days in callus induction and proliferation medium for all lines of *C. arabica*. 'Oeiras' exhibited the highest mean value of 4.20 responsive explants at 15 days, followed by 'Caturra' with 3.82, and 'Catuaí Vermelho' with 2.17. The mean number increased at 30 days for 'Oeiras', 'Caturra' and 'Catuaí Vermelho', which showed, respectively, 4.32, 3.98 and 2.32 responsive explants. The mean number remained stable until 75 days for 'Oeiras'. In 'Caturra', the induction of responsive explants increased at 45 days and 60 days, showing a mean value of 4.00 and 4.02, respectively, remaining stable at 75 days. For 'Catuaí Vermelho', the mean number remained constant until 45 days, increasing for 2.36, respectively, at 60 and 75 days. At 90 days, the mean number of 'Oeiras', 'Caturra' and 'Catuaí Vermelho' responsive explants respectively increased for 4.35, 4.04 and 2.37. As seen, 'Oeiras' had the highest mean number of responsive explants in all evaluated periods, followed by 'Caturra' and 'Catuaí Vermelho' (Figure 1A). The time did not influence the rate of calli induction in 'Oeiras'. Differently, 'Catuaí Vermelho' and 'Caturra' calli induction increased over time. Visually, the callus proliferation gradually increased at 30 days, becoming stable at 90 days. The friable calli from all *C. arabica* lines showed pale-yellow coloration (Figure 1B).

During SE regeneration, *C. arabica* lines showed distinct MCSE mean values over time. The 'Catuaí Vermelho' friable calli presented the first MCSE at 30 days with a mean value of 0.03 MCSE, while the first MCSE of 'Caturra' were observed at 60 days with a mean value of 0.02 (Figure 2A). 'Catuaí Vermelho' SE regeneration increased after 30 days, showing mean values of 2.11, 4.78, 7.39, 10.44 and 12.25 MCSE per callus at 60, 90, 120, 150 and 180 days, respectively. 'Caturra' SE regeneration increased after 60 days, showing mean values of 0.05, 0.11, 0.22 and 0.24 MCSE per callus at 90, 120, 150 and 180 days, respectively (Figure 2). 'Catuaí Vermelho' regenerated the highest MCSE mean number in all evaluated

times, differing statistically from 'Caturra' at 60, 90, 120, 150 and 180 days (Figure 2A). Therefore, ISE was established for 'Catuaí Vermelho' and 'Caturra' (Figure 3A,B). Globular, cordiform, torpedo and cotyledonary SE were regenerated, evidencing an asynchronous response during ISE. SE regeneration continued for over 180 days for 'Catuaí Vermelho' and 'Caturra', with seedling recovery potential over several months. ISE was not established for 'Oeiras', since the friable calli did not acquire embryogenic competence (Figure 3C).

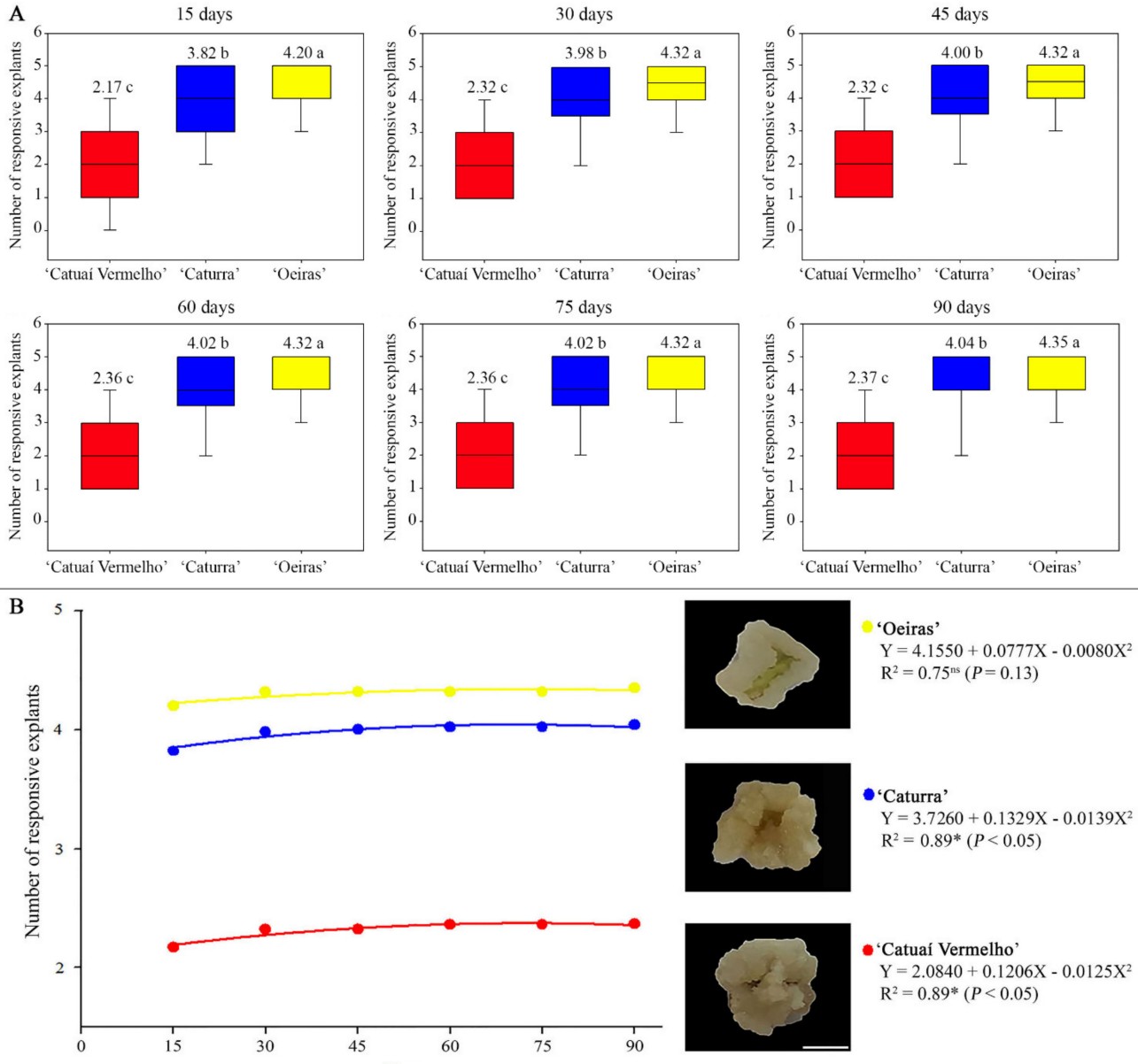

**Figure 1.** Mean number of responsive explants of 'Catuaí Vermelho', 'Caturra' and 'Oeiras' in callus induction and proliferation medium over 90 days. (**A**) The *C. arabica* lines exhibited distinct values of responsive explants over 90 days. (**B**) The adjusted model was significant ($p < 0.05$) by the quadratic polynomial regression for 'Catuaí Vermelho'. All calli exhibited a pale-yellow color and friable appearance. Mean values followed by the same letter do not present a significant difference. Bar = 2 mm.

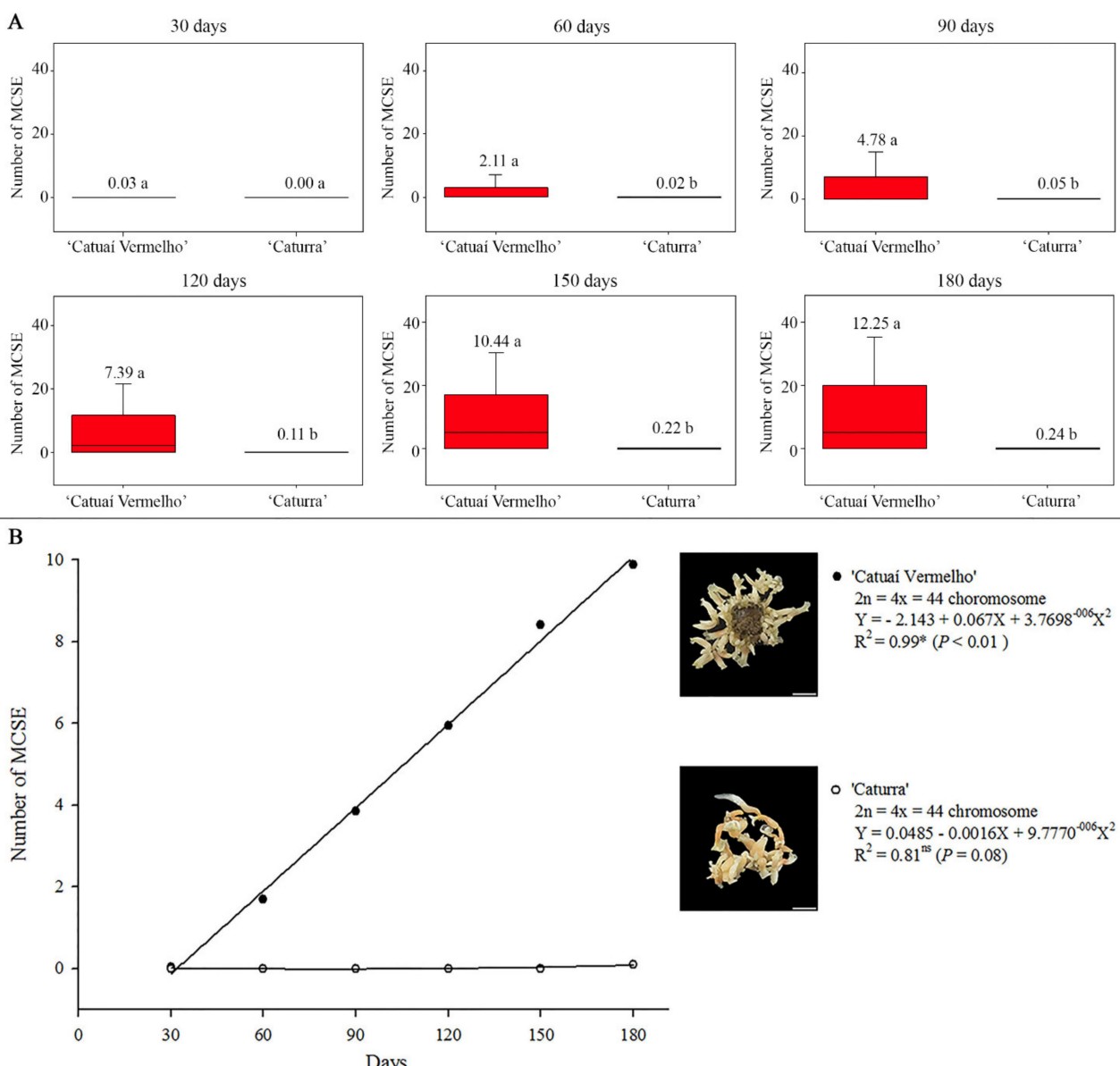

**Figure 2.** MCSE mean number for 'Catuaí Vermelho' and 'Caturra' in SE regeneration medium for 180 days. (**A**) Box plots show that 'Catuaí Vermelho' and 'Caturra' exhibited distinct MCSE mean numbers. (**B**) The adjusted model was significant ($p < 0.01$) by the quadratic polynomial regression for 'Catuaí Vermelho''. The regenerated SE exhibited different stages of globular, cordiform, torpedo and cotyledonary development. Mean values followed by the same letter do not present a significant difference. Bar = 2 mm.

### 3.2. Genomic Methylated Cytosine Level

'Catuaí Vermelho', 'Caturra' and 'Oeiras' friable calli presented distinct mean values of genomic methylated cytosine level at 60 and 90 days. Friable calli from 'Oeiras' and 'Caturra' exhibited mean values of methylated cytosine equivalent to 34.34% and 38.70%, respectively, at 60 days, statistically differing from 'Catuaí Vermelho' that showed a mean value of 20.73% (Figure 4A). At 90 days, friable callus from 'Caturra' exhibited a mean value of 53.40% of methylated cytosine, statistically higher than 'Oeiras' and 'Catuaí Vermelho' that had mean values of 33.51% and 30.80%, respectively (Figure 4B). Methylated cytosine levels gradually increased over time for 'Catuaí Vermelho' (20.73% at 60 days and 30.79% at 90 days, Figure 4C) and 'Caturra' (38.70% at 60 days and 53.40% at 90 days, Figure 4D), while for 'Oeiras', the mean values (34.34% at 60 days and 33.51% at 90 days) did not differ significantly (Figure 4E).

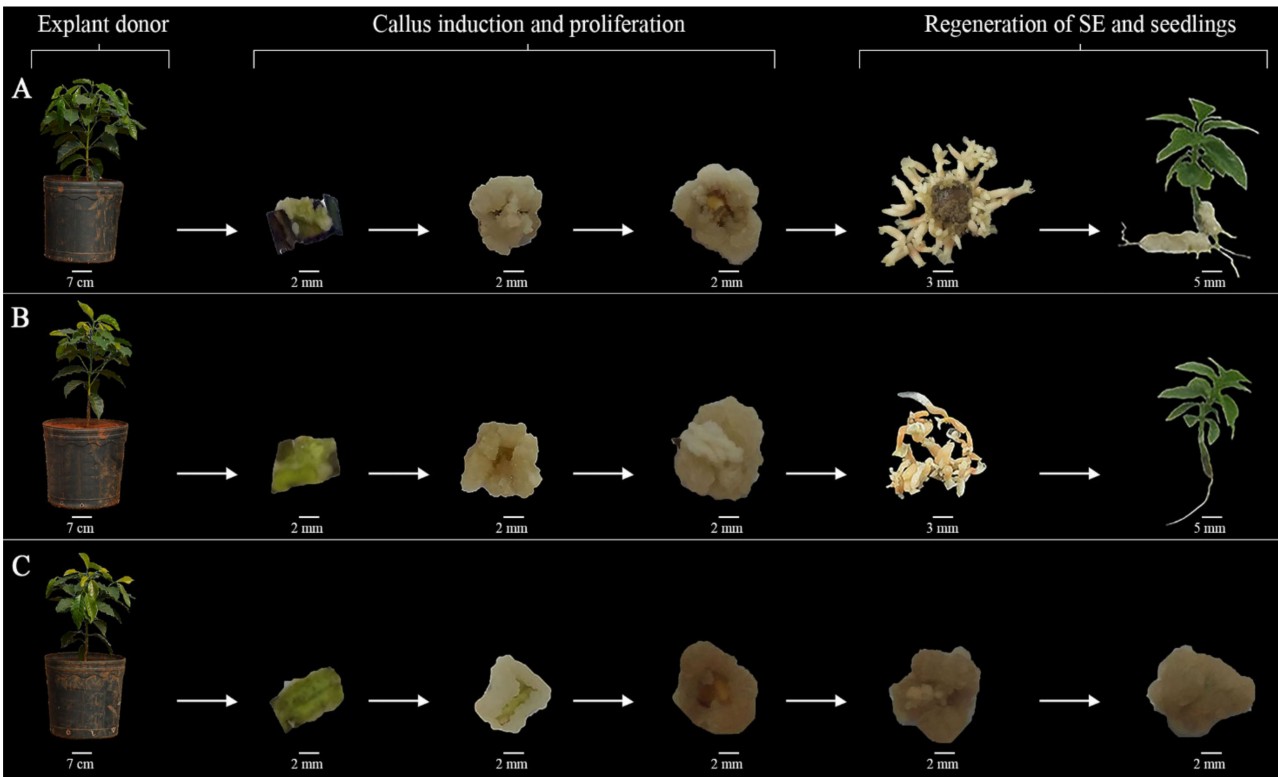

**Figure 3.** ISE response of the *C. arabica* lines during in vitro culture. ISE response was evaluated in three stages: induction and proliferation of friable calli; and regeneration of SE and seedlings. Leaf explants responded the induction and proliferation conditions, resulting in friable callus for the three lines (**A–C**). ISE was established for 'Catuaí Vermelho' and 'Caturra', acquiring embryogenic competence and regenerating plants (**A,B**). ISE was not stablished for 'Oeiras', remaining in friable callus stage.

Embryogenic calli of 'Catuaí Vermelho' and 'Caturra' presented different methylated cytosine mean values (Figure 4F). The highest level of methylated cytosine was observed in embryogenic calli of 'Caturra' with 54.09%, while the embryogenic calli of 'Catuaí Vermelho' exhibited a mean value of 43.35% (Figure 4F). 'Catuaí Vermelho' had a 12.56% (30.79 to 43.35%) increase of methylated cytosine while 'Caturra' showed only a 0.69% increase (53.40 to 54.09%) (Figure 4B,F). In general, the step of induction and proliferation of friable calli was characterized by lower values of methylated cytosine, and the second step involving SE regeneration was marked by higher values of methylated cytosine.

### 3.3. Ploidy Level Stability

All regenerated 'Catuaí Vermelho' plantlets exhibited mean values of 2C = 2.62 pg $\pm$ 0.027 and chromosomal number of 2n = 4x = 44. The in vitro regenerated plantlets of 'Caturra' showed 2C = 2.60 pg $\pm$ 0.021 and the same chromosome number. In 'Oeiras', it was not possible to carry out these analyses, since it was not established in vitro. Therefore, based on these data on 2n chromosome number, ploidy level and nuclear 2C value, no karyotypic variations were observed between explant donors and plantlets regenerated in vitro from 'Catuaí Vermelho' and 'Caturra' (Figure 5).

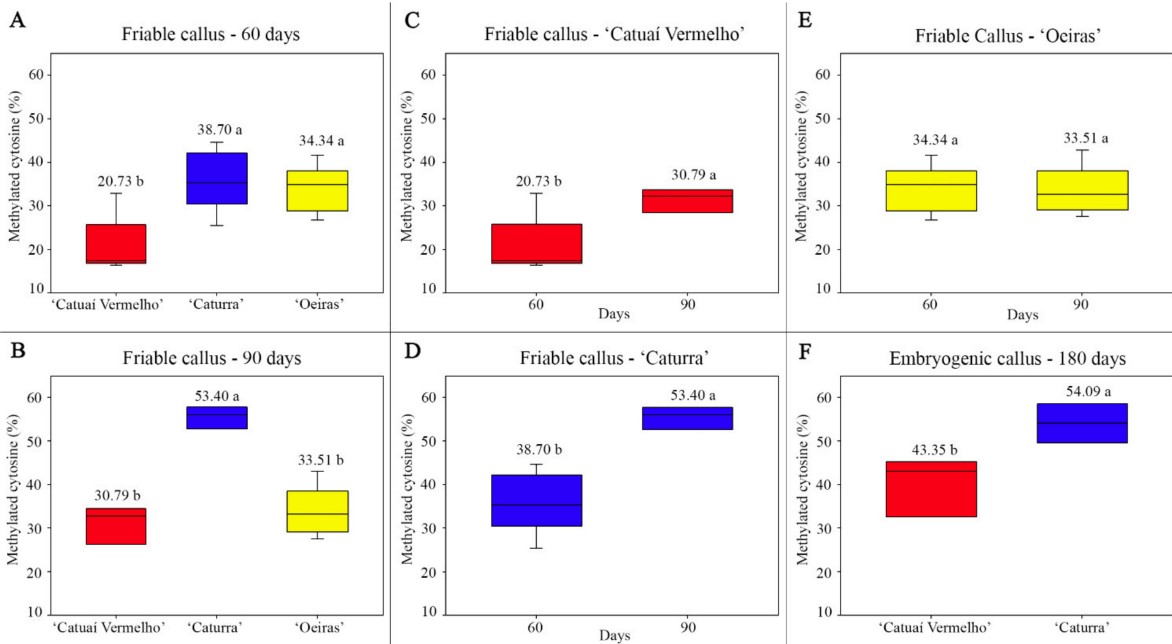

**Figure 4.** Levels of global methylation of genomic DNA in *C. arabica* lines during ISE. (**A**) Comparison of the methylated cytosine level between the friable callus of 'Catuaí Vermelho', 'Caturra' and 'Oeiras' at 60 days. (**B**) Comparison of the methylated cytosine level between the friable callus of 'Catuaí Vermelho', 'Caturra' and 'Oeiras' at 90 days. (**C**) Comparison of the methylated cytosine level in friable callus of 'Catuaí Vermelho' at 60 and 90 days. (**D**) Comparison of the methylated cytosine level in friable callus of 'Caturra' at 60 and 90 days. (**E**) Comparison of the methylated cytosine level in friable callus of 'Oeiras' at 60 and 90 days. (**F**) Comparison of the methylated cytosine level between the embryogenic callus of 'Catuaí Vermelho' and 'Caturra' at 180 days. Mean values followed by the same letter do not present a significant difference.

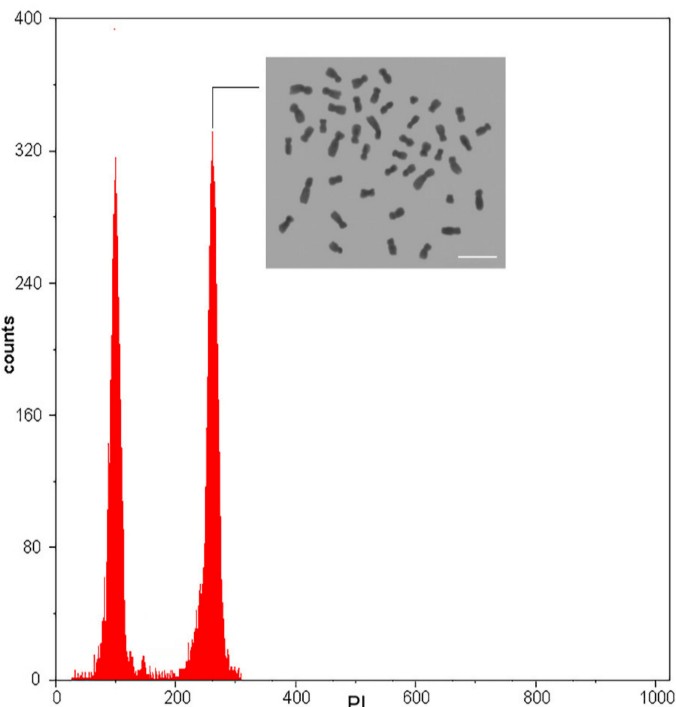

**Figure 5.** Nuclear DNA content and ploidy confirmation/chromosomal number of *C. arabica* seedlings regenerated in vitro. Representative histogram showing G0/G1 peaks of intact *S. lycopersicum* nuclei in channel 100 and 'Catuaí Vermelho' or 'Caturra' in channel 262 and ploidy/chromosome number confirmation 2n = 4x = 44. Bar = 5 μm.

## 4. Discussion

In vitro procedures were developed under the same conditions to compare and evaluate *C. arabica* lines 'Catuaí Vermelho', 'Caturra' and 'Oeiras' in relation to ISE response. In addition, it was possible to determine, compare and associate genomic methylated cytosine level in the *C. arabica* lines during ISE responses and to evaluate the karyotype stability of in vitro regenerated plantlets. The three tested *C. arabica* lines differed in their in vitro response (mean number of responsive leaf explants, mean number of MCSE and response time) and in their global methylated cytosine levels. ISE was established for 'Catuaí Vermelho' and 'Caturra' based on an improved and reproducible procedure proposed by [4,5], involving the callus induction and proliferation, followed by the SE and plantlets regeneration. However, the ISE procedure (in vitro conditions) was not effective for 'Oeiras' since there were no SE recovered, the development remained in the callus phase. For ISE establishment, differentiated plant cells must return to their undifferentiated state and regain totipotency, giving rise to calli and later competence acquisition to regenerate SE [15]. According to [39], the process of acquiring embryogenic competence by somatic cells should involve the reprogramming of gene expression patterns, as well as changes in physiology and morphology. These changes reflect dedifferentiation, activation of cell division, and a change in cell fate through down-regulation of some genes operating in differentiated cells and up-regulation of genes needed for transition [40]. However, Campos et al. [3] and Mendes et al. [41] suggest that plant cells do not dedifferentiate, but that callus is formed from pre-existing stem cells. These cells maintain their totipotency throughout plant development and, under the adequate stimulus, multiply and differentiate recovering the SE.

During the callus induction and proliferation, 'Oeiras' leaf explants were the most responsive, providing friable calli in the relatively short time of one month, followed by 'Caturra' and 'Catuaí Vermelho'. In the second ISE step (acquisition of competence, determination and differentiation of friable callus cells and regeneration of SE), a higher mean number of MCSE was observed for 'Catuaí Vermelho' compared to 'Caturra', which presented the highest mean number of responsive explants. Therefore, our data show differences between all three *C. arabica* lines, suggesting that the in vitro response is dependent and influenced by the genotype. The 'Caturra' lines originated from a natural mutation of the 'Bourbon' line; the breeding method was mass selection based on the phenotype without progeny testing [41]. The 'Oeiras' line was developed by the pedigree method from the hybrid 'CIFC HW 26/5', which was obtained from a crossing between 'Caturra Vermelho' (CIFC 19/1) and 'Híbrido de Timor CIFC 832/1' [42]. The 'Catuaí Vermelho' originated from a recombination of the 'Caturra Amarelo' and 'Mundo Novo' lines [41]. Therefore, all *C. arabica* lines here under study originated and were selected by different breeding methods and have a restricted but different genetic base, which may have influenced the in vitro response. Corroborating, other studies also showed that in vitro response in *C. arabica* is genotype dependent and influenced by the in vitro environment [4,42,43], evaluating the in vitro ISE responses of the commercial *C. arabica* 'Caturra Rojo' and the wild type *C. arabica* of the Ethiopian (ET 25.1, ET 20.1, ET 1.1, ET 12.4, ET 12.5, KF 2.1 and KF 6.3), observed that only KF 2.1, ET 25.1, ET 12.5 and ET 1 regenerated SE and plantlets. In addition, plant growth regulators and endogenous phytohormones also influence somatic embryogenesis in *Coffea* [4,23].

Our data show that the in vitro tissue culture condition alters the global methylated cytosine levels since there was a gradual increase over time in friable calli 'Catuaí Vermelho' (20.73% at 60 days and 30.79% at 90 days) and 'Caturra' (38.70% at 60 days and 53.40% at 90 days). Variations in methylated cytosine levels were also observed between *C. arabica* lines during the SE regeneration. The increase of the methylated cytosine levels observed in in vitro established *C. arabica* lines is associated with a chromatin remodeling from an euchromatic to a heterochromatic state. The callus formation is characterized by cell proliferation influenced by in vitro conditions. Cell proliferation is marked by euchromatin status. After transferring to regeneration medium, the callus cells are exposed to the

new and different in vitro conditions. This change induces the heterochromatin status by methylation of the cytosine. Cytosine methylation is a conserved epigenetic modification that plays an important role in chromatin remodeling [44] and transcriptional control of gene expression, influencing the somatic embryogenesis establishment [22,45]. In addition, chemical changes of the histone amino acids probably occur, which should be investigated in further studies.

Gene regulation mechanisms have been the subject of studies on several plant species of agronomic interest, in order to investigate and elucidate morphogenetic events in vitro [44–46]. According to [22], epigenetic changes in somatic embryogenic tissues of *C. canephora* are controlled by histone modifications and methylated cytosine. As observed in this study, an increase of methylated cytosine may be associated with the regeneration and maturation of the SE in *C. arabica*, corroborating with the previous studies with the *Coffea* genus [22,23,26,46]. According to [44], cytosine methylation is a complex and variable chemical change. In addition, the epigenetic marks can be heritable over several generations or can be induced by in vitro environmental conditions. Therefore, the different values of methylated cytosine observed here and their variations during the ISE may be related to the improvement methods that gave rise to these *C. arabica* lines and/or the adaptation processes of the cells to environmental in vitro conditions. Cell changes to the in vitro environment of ISE are accompanied by chromatin remodeling, which influences the gene expression [47,48]. In addition, cytosine methylation is associated with genome stability by suppressing the transposition of the mobile DNA elements in plants [49].

The occurrence of somaclonal variation is undesirable in plant tissue culture, as a high level of somaclonal variation can ruin a possible genotype. According to [28], low rates of somaclonal variation were identified in *C. arabica* genotypes using molecular markers. Corroborating, [27] noted that 99% of *C. arabica* trees regenerated in vitro are in compliance with the explant donor plant and that phenotypic variants are induced by aneuploidy, showing that this variation occurs during in vitro propagation. However, genetic variations (euploidy and aneuploidy) were not observed in the present study between in vitro regenerated plantlets in relation to leaf explant donors. Flow cytometry analysis showed that the regenerated plantlets had mean 2C DNA values (2C = 2.62 pg) and ploidy level identical to those of the explant donor plant. In addition, the cytogenetic approach confirmed that regenerated plantlets of 'Catuaí Vermelho' and 'Caturra' remain tetraploid, with 2n = 4x = 44 chromosomes. Therefore, our data showed that the ISE protocol is reproducible, viable and safe since it did not promote chromosomal alterations.

## 5. Conclusions

*C. arabica* lines exhibited different responses in vitro during dedifferentiation, cellular competence, and regeneration of SE and plantlets, exhibiting the same ploidy level of the explant donor plant. The differences were correlated with methylated cytosine level since it was shown to be dynamic and variable during ISE. In addition, *C. arabica* ISE was associated with cytosine hypermethylation. Based on this study, further approaches are needed to measure the methylated cytosine level of the specific genes during *Coffea* ISE, as well as the outcome of this epigenetic change in gene expression. In addition, the levels of the endogenous hormones should be measured during in vitro development in an attempt to clarify the different responses observed.

**Supplementary Materials:** The following are available online at https://www.mdpi.com/article/10.3390/f12111536/s1, Supplement 1: Tissue culture medium used for ISE establishment from leaf explants of *C. arabica* 'Catuaí Vermelho', 'Caturra' and 'Oeiras'.

**Author Contributions:** J.P.d.M.O. and W.R.C. designed this work, conducted all experiments, and wrote the manuscript. N.A.S. co-oriented and assisted in the experiments of this work. A.F. performed the statistical analysis. All authors have read and agreed to the published version of the manuscript.

**Funding:** This research was funded by Conselho Nacional de Desenvolvimento Científico e Tecnológico (CNPq, Brasília—DF, Brazil, grant: 443801/2014-2), the Fundação de Amparo à Pesquisa do Espírito Santo (FAPES, Vitória—ES, Brazil, grants: 65942604/2014 and 82/2017), and the Coordenação de Aperfeiçoamento de Pessoal de Nível Superior (CAPES, Brasília—DF, Brazil) for financial support.

**Institutional Review Board Statement:** Not applicable.

**Informed Consent Statement:** Not applicable.

**Data Availability Statement:** Not applicable.

**Conflicts of Interest:** The authors declare no conflict of interest. The funders had no role in the design of the study; in the collection, analyses, or interpretation of data; in the writing of the manuscript; or in the decision to publish the results.

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
