# Peer review of "Genomic Methylated Cytosine Level during the Dedifferentiation and Cellular Competence in Coffea arabica Lines: Insights about the Different In Vitro Responses"

_forests, doi:10.3390/f12111536_

Round 1

Reviewer 1 Report

Reviewer report:

Coffee is one of the most important commodities cultivated worldwide with a great economic impact in many countries. Conventional breeding requires more than 30 years to have superior genotypes in the market. The necessity of biotechnological propagation methods such as in vitro techniques should be taken into account as they would be a powerful tool to implement coffee breeding programs. In this work authors have evaluated the genotypic influence on the morphogenic response in vitro  and the role of global cytosine methylation in three lines of Coffea arabica. For these purposes they have established and compared indirect somatic embryogenesis in the three lines under the same in vitro conditions, verified the influence of the global levels of methylated cytosine during dedifferentiation and cellular competence, and checked the stability of regenerated plantlets in vitro.

It is well known that indirect somatic embryogenesis can lead to the appearance of somaclonal variation and it’s a positive point that the authors evaluated the stability of the regenerated plants. However, the authors give a slight explanation about the role of cytosine methylation.

After a thorough reading my recommendation is that the authors should revise some aspects of the manuscript. Bellow the authors will find some specific comments and suggestions.  

 Abstract:

  •  Be consistent with the spelling with Latin expressions: in vitro, species names, etc, should be written in cursive.
  • Line 13: “Have shown” is more correct.
  • Line 18: “Leaf explants were inoculated…”  should be changed by “Leaf explants were cultured…”

 Introduction:

  •  Line 57: The reference Nic-Can et al. 2015 doesn´t appear in the reference list
  • Line 73: “… the evaluating of the genomic….” should be changed by “…the evaluation of the genomic…”

Material and Methods:

  • Line 109: “Leaf explants were inoculated…”  should be changed by “Leaf explants were cultured…”
  • Line 137: Oliveira et al. (2021)

 Results:

 In general, this part should be considered to be written more carefully.

When the authors talk about de IES establishment it should be pointed out that the increases in the values of the responsive explants, in general, are quite low.

The authors mention that the time didn´t influence the rate in calli induction in Oerias and Caturra, whereas in Catuaí  the rate increased overtime (Lines 206-207). However, in Fig. 1B,  p-value for Caturra  is lower than 0.05 so it seems that for this line time influences the rate of calli induction.

During SE regeneration, it is pointed out by the authors that  Catuaí  regenerated the highest MSCE mean number in all evaluated times, differing statistically from Caturra after 30 days (Lines 223-224). After 30 days, Fig 2A indicates that there are not significant differences between Catuaí and Caturra, but differences are observed after 60, 90, 120, 150 and 180 days.

The sentence in line 224 “ No SE Oeiras  was recovered for up to 180 days”  is in a way repeated in lines 229-230 “ ISE was not established for Oeiras…”. Merge these two sentences in just one.

Can the authors detect morphological differences between friable calli that acquire embryogenic competence from those that don´t acquire it? It could be very interesting to do some histological studies in order to characterize the morphology of the calli, maybe it could explain why some of them develop the embryogenic competence.

Genomic methylated cytosine level results section presents confusing redaction. There are mismatches between the data along the text and the data that appears in the graphs in Figs. 4A and 4B for Caturra: in line 247 appears 35.35% vs 38.70% in Fig 4A;  in line 249 appears 50.40% vs 53.40% in Fig 4B. Also the authors should explain clearer what they want to express in the sentence from line 267 to 269.

Authors have measured the genomic methylated cytosine level in friable calli and in embryogenic calli, maybe it is interesting to measure this levels in the initial explant (leaves), before callus induction.  

Discussion:

Authors must discuss with more detail what happens once embryogenic competence  is acquired and the increase of methylation levels, try to relate it. Also, any explanation about why the most MSCE productive line (Catuaí) presents lower methylated cytosine levels compared to Caturra is missing.

 Figures:

 In general, footnotes should be rewritten in Figs 1, 2 and 4. They should indicate what they represent (specially in Figs 1 and 2), the meaning of the letters (i.e. significant differences), the dots, etc,.

Footnote for Fig 3 should be completely rewritten too, it is suggested “ Behavior of the C. arabica lines during the establishment of the ISE……and regeneration of SE and seedlings. Leaf explants responded for induction and proliferation conditions, resulting in friable callus for the three lines (A-C). ISE was established for Catuaí and Caturra, acquiring embryogenic competence and regenerating plants (A, B). ISE was not stablished for Oeiras, remaining in friable callus stage”. 

The authors must consider to use bigger size for letters and numbers in all the figures, they are too small and difficult to see. Also Figure 3 is too dark.

 Conclusions:

 According to the results obtained, which are the new approaches needed? Can the authors specify more this aspect?

 References:

There are references that are listed in the reference list but doesn´t appear along the text: references nº 7, 17 and 45.

Overall, the manuscript has some shortcomings. It is suggested a requirement for revision.

Author Response

It is with great pleasure that we are resubmitting to Forests the article “Genomic methylated cytosine level during the dedifferentiation and cellular competence in Coffea arabica lines: insights about the different in vitro responses".

All criticisms were considered and the manuscript modified according to Reviewers’ suggestions. According to Forests rules, all changes were highlighted in the revised manuscript, using the fluorescent highlighter “yellow”. Thanks for being so supportive towards our manuscript. Hope to hearing from you soon!

REVIEWER 1

Coffee is one of the most important commodities cultivated worldwide with a great economic impact in many countries. Conventional breeding requires more than 30 years to have superior genotypes in the market. The necessity of biotechnological propagation methods such as in vitro techniques should be taken into account as they would be a powerful tool to implement coffee breeding programs. In this work authors have evaluated the genotypic influence on the morphogenic response in vitro  and the role of global cytosine methylation in three lines of Coffea arabica. For these purposes they have established and compared indirect somatic embryogenesis in the three lines under the same in vitro conditions, verified the influence of the global levels of methylated cytosine during dedifferentiation and cellular competence, and checked the stability of regenerated plantlets in vitro.

It is well known that indirect somatic embryogenesis can lead to the appearance of somaclonal variation and it’s a positive point that the authors evaluated the stability of the regenerated plants. However, the authors give a slight explanation about the role of cytosine methylation.

After a thorough reading my recommendation is that the authors should revise some aspects of the manuscript. Bellow the authors will find some specific comments and suggestions.

Abstract:

Be consistent with the spelling with Latin expressions: in vitro, species names, etc, should be written in cursive.

Line 13: “Have shown” is more correct.

Line 18: “Leaf explants were inoculated…”  should be changed by “Leaf explants were cultured…”

Authors’ comments:

The manuscript was modified according to all suggestions.

Introduction:

Line 57: The reference Nic-Can et al. 2015 doesn´t appear in the reference list

Line 73: “… the evaluating of the genomic….” should be changed by “…the evaluation of the genomic…”

Authors’ comments:

The manuscript was modified according to all suggestions.

Material and Methods:

Line 109: “Leaf explants were inoculated…”  should be changed by “Leaf explants were cultured…”

Line 137: Oliveira et al. (2021)

Authors’ comments:

The manuscript was modified according to all suggestions.

Results:

In general, this part should be considered to be written more carefully.

When the authors talk about de IES establishment it should be pointed out that the increases in the values of the responsive explants, in general, are quite low.

The authors mention that the time didn´t influence the rate in calli induction in Oerias and Caturra, whereas in Catuaí  the rate increased overtime (Lines 206-207). However, in Fig. 1B,  p-value for Caturra  is lower than 0.05 so it seems that for this line time influences the rate of calli induction.

During SE regeneration, it is pointed out by the authors that  Catuaí  regenerated the highest MSCE mean number in all evaluated times, differing statistically from Caturra after 30 days (Lines 223-224). After 30 days, Fig 2A indicates that there are not significant differences between Catuaí and Caturra, but differences are observed after 60, 90, 120, 150 and 180 days.

The sentence in line 224 “ No SE Oeiras  was recovered for up to 180 days”  is in a way repeated in lines 229-230 “ ISE was not established for Oeiras…”. Merge these two sentences in just one.

Genomic methylated cytosine level results section presents confusing redaction. There are mismatches between the data along the text and the data that appears in the graphs in Figs. 4A and 4B for Caturra: in line 247 appears 35.35% vs 38.70% in Fig 4A;  in line 249 appears 50.40% vs 53.40% in Fig 4B. Also the authors should explain clearer what they want to express in the sentence from line 267 to 269

Authors’ comments: The manuscript was modified according to all suggestions.

Can the authors detect morphological differences between friable calli that acquire embryogenic competence from those that don´t acquire it? It could be very interesting to do some histological studies in order to characterize the morphology of the calli, maybe it could explain why some of them develop the embryogenic competence.

Authors have measured the genomic methylated cytosine level in friable calli and in embryogenic calli, maybe it is interesting to measure this levels in the initial explant (leaves), before callus induction.

Authors’ comments: Cytological differences between cells of the calli, as well as morphological differences between the calli, have been identified and showed mainly by cytological approaches, including electron microscopy (MEV or MET). We agree with the suggestion, but the cytological characterization is other phenotype data about the ISE, as well as the somatic embryo regeneration. For this, we investigated the epigenetic influence under ISE in Coffea.

Discussion:

Authors must discuss with more detail what happens once embryogenic competence  is acquired and the increase of methylation levels, try to relate it. Also, any explanation about why the most MSCE productive line (Catuaí) presents lower methylated cytosine levels compared to Caturra is missing.

Authors’ comments: The manuscript was modified according to all suggestions. The Discussion section was changed.

Figures:

In general, footnotes should be rewritten in Figs 1, 2 and 4. They should indicate what they represent (specially in Figs 1 and 2), the meaning of the letters (i.e. significant differences), the dots, etc,.

Footnote for Fig 3 should be completely rewritten too, it is suggested “ Behavior of the C. arabica lines during the establishment of the ISE……and regeneration of SE and seedlings. Leaf explants responded for induction and proliferation conditions, resulting in friable callus for the three lines (A-C). ISE was established for Catuaí and Caturra, acquiring embryogenic competence and regenerating plants (A, B). ISE was not stablished for Oeiras, remaining in friable callus stage”.

The authors must consider to use bigger size for letters and numbers in all the figures, they are too small and difficult to see. Also Figure 3 is too dark.

Authors’ comments: The manuscript was modified according to all suggestions.

Conclusions:

According to the results obtained, which are the new approaches needed? Can the authors specify more this aspect?

Authors’ comments: According to this comment, we modified the Conclusion section to appoint the further approaches needed.

References:

There are references that are listed in the reference list but doesn´t appear along the text: references nº 7, 17 and 45.

Authors’ comments: The manuscript was modified according to all suggestions.

Reviewer 2 Report

The results are preliminary and only describe the phenomena. The obtained results do not allow for the identification of any mechanisms participating in the dedifferentiation. Responsive callus and SE regenerative in three varieties of Coffea Arabica were examined in publication. This part of the research is well done and shows clear results. Next, attempts were made to determine the changes in the 5mC level during de-differentiation and indirect somatic embriogenesis. However, only changes in total m5C during at 60 and 90 days were shown. These changes do not correlate with the effectiveness of somatic embryogenesis in the varieties studied. The presented studies do not explain the role of m5C in dedifferentiation. Whether 5mC inhibition affects somatic embryogenesis (5-AzaC) has not been investigated. To this end, the relative expression of transcription factors (TFs) of the documented regulatory function in SE induction including LEC1 (LEAFY COTYLEDON1), LEC2 (LEAFY COTYLEDON2) and BBM (BBM BABY BOOM) in callus treated and untreated 5-AzaC must be determined. The study of 5mC function should include DNA sequencing after bisulfite reaction. The presented results in no explain the role of 5mC in somatic embryogenesis in Coffea Arabica. Then, the occurrence of somaclonal variation was investigated. In my opinion, the results are poorly presented. Three cultivars were not tested. There was no comparison of nuclear DNA content between leaf explants and callus or SE. Cytogenetic studies document only one of the three varieties. The results of the work document somatic embryogenesis well, therefore it may be more suitable for journals on plant biotechnology. However, the lack of any mechanisms of 5mC operation on dedifferentiation, in my opinion, does not quite fit in Forest.

Author Response

It is with great pleasure that we are resubmitting to Forests the article “Genomic methylated cytosine level during the dedifferentiation and cellular competence in Coffea arabica lines: insights about the different in vitro responses".

All criticisms were considered and the manuscript modified according to Reviewers’ suggestions. According to Forests rules, all changes were highlighted in the revised manuscript, using the fluorescent highlighter “yellow”. Thanks for being so supportive towards our manuscript. Hope to hearing from you soon! 

REVIEWER 2

The results are preliminary and only describe the phenomena. The obtained results do not allow for the identification of any mechanisms participating in the dedifferentiation. Responsive callus and SE regenerative in three varieties of Coffea Arabica were examined in publication. This part of the research is well done and shows clear results. Next, attempts were made to determine the changes in the 5mC level during de-differentiation and indirect somatic embriogenesis. However, only changes in total m5C during at 60 and 90 days were shown. These changes do not correlate with the effectiveness of somatic embryogenesis in the varieties studied. The presented studies do not explain the role of m5C in dedifferentiation. Whether 5mC inhibition affects somatic embryogenesis (5-AzaC) has not been investigated. To this end, the relative expression of transcription factors (TFs) of the documented regulatory function in SE induction including LEC1 (LEAFY COTYLEDON1), LEC2 (LEAFY COTYLEDON2) and BBM (BBM BABY BOOM) in callus treated and untreated 5-AzaC must be determined. The study of 5mC function should include DNA sequencing after bisulfite reaction. The presented results in no explain the role of 5mC in somatic embryogenesis in Coffea Arabica. Then, the occurrence of somaclonal variation was investigated. In my opinion, the results are poorly presented. Three cultivars were not tested. There was no comparison of nuclear DNA content between leaf explants and callus or SE. Cytogenetic studies document only one of the three varieties. The results of the work document somatic embryogenesis well, therefore it may be more suitable for journals on plant biotechnology. However, the lack of any mechanisms of 5mC operation on dedifferentiation, in my opinion, does not quite fit in Forest.

Authors’ comments: We disagree of the comment about the role of the 5-mC because we compared the genomic 5-mC level between responsive and non-responsive calli based on ISE response. We know about 5-AzaC, but this compound has been used to show its epigenetic effect mainly when somatic embryos did not regenerate. Sure, clearly, that's not the reality for Coffea. The measurement of the 5-mC of the regulatory portions of some genes has been accomplished, and we have performed this analysis without 5-AzaC treatment due to our aims, used Coffea species and genetic context. According to our manuscript, all regenerated C. arabica plantlets were investigated in related to ploidy stability. As the resulted is the same, we showed representative data because it is not necessary to show the same result.

Reviewer 3 Report

Joao Paulo de Morais Oliveira and Colleagues presented a paper investigating the level of global genomic DNA methylation in Coffea arabica plants. The Authors focused on three aspects: (1) establish the indirect somatic embryogenesis in C. arabica, (b) associate the distinct responses to the global level of DNA methylation, (c) check the stability of ploidy in regenerants.

All of them seem vital and relevant for readers interested in in vitro cultures; however, the authors did not avoid shortcomings. Main note, the manuscript needs linguistic smoothing as some sentences do not fully convey the statement's meaning. However, it seems that improving the language would significantly increase the value of the manuscript.

Specific comments:

Abstract

  • “From these somatic embryos, plantlets were recovered, exhibiting the same 2C value 27 and chromosome number than plant donors”

….than?... There seems to be an inaccuracy in this sentence; please correct it.

Introduction

  • “Indirect somatic embryogenesis (ISE) is a morphogenetic pathway that consists of the cultivation of plant tissue segments in sterile and specific culture medium, in the originating callus and, later, in the somatic embryos (SE) and plantlets.”

This sentence is not entirely clear. It needs rewording. ….(ISE)..that consists of… in the originating callus????...

  • “The somaclonal variation is associated to genomic (nucleus, mitochondria and plastid) and/or epigenomic changes, which can result in phenotypic variations of the regenerated plantlets in relation to the plant explant donors – called genetic fidelity loss (de Oliveira et al. 2019).”

….plant explant donors… or as previously line 53 …explant donor plant…??? It would be good if the Authors adopted one convention to describe donor plants (source of explants) and stuck to this version throughout the manuscript. Therefore, please consider this comment throughout the text.

  • “These variations….????

Authors write about somaclonal variation, so I guess it should be “This variation…”

I would like to know whether the examples the Authors give from the literature actually refer to somaclonal variation (SV) or perhaps to tissue culture-induced variation (TCIV)? From the definition of somaclonal variation, it is variation induced in vitro and inherited in the generative progeny of regenerants and should be analyzed not in regenerants but in the generative progeny of regenerants. Very often, these terms are used interchangeably, probably not quite rightly.  Hence my question is if the quoted examples concern SV or TCIV?

  • “Somaclonal variations can be verified…. “

In this sentence, the Authors want to present methods to study variation induced in plant tissue culture. However, there is no consistency here; while flow cytometry is a technique, the other cited examples are not. Authors could write,e.g., with molecular marker techniques, or at what level this variability can be identified, e.g., at the level of DNA sequence or DNA or histone methylation. Could Authors choose a convention for their description and stick to it? “In this study, we used three lines of C. arabica, 'Catuaí Vermelho', 'Caturra' and 'Oeiras', which originated from genetic recombination, mass selection and pedigree…?

Could the Authors specify more precisely what "pedigree" means in this case? Perhaps it would be worth adding 1-2 sentences of explanation.

Statistical analysis

I am curious why the Authors have such a large discrepancy in repetitions of 8, 9, 20, or 6 and 20?

Results

The description of statistical analyzes of results lacks whether there were significant outliers in the groups of the independent variable in terms of the dependent variable, whether the dependent variable was approximately normally distributed for each group, and whether the homogeneity of variances was preserved? It would be good to provide the results of the appropriate tests checking the above issues.

line 207: typo  …'Catuaí Vemelho…

lines 212-214: Is a hyphen before the figure caption needed in all figures? I would move the phrase “The highest mean number of responsive explants was observed in 'Oeiras', followed by 'Caturra' and 'Catuaí Vermelho' for all evaluated times.” to the description of the results rather than leaving it in the figure caption.

line 232:…As in the comment above, in the figure caption, I would leave only the relevant description concerning the graph and pictures and move the descriptions to the results text.

line 238:… “Leaf explants responded to the means of induction and proliferation, resulting in friable callus. However, friable calli did not acquire embryogenic competence and remained in a stage of callus.3.1. Subsection.” I would move this text to the description of the results.

Discussion

line 331: “Our data show that the in vitro tissue culture condition alters the methylation patterns, …” I am not entirely convinced that the Authors should use the phrase "methylation pattern"; after all, they investigated the global level of methylation in genomic DNA rather than the location of methylated cytosines, e.g., in specific CG, CHG, or CHH sequences. Therefore, I would be careful with such a statement.

line 337: “….from a eucromatic to a…” typo

Lines 346-349: This high degree of cytosine methylation in plants is attributed to the fact that, in plants, methylation can be present in three CG islands, CHG and CHH, where H can be any deoxynucleotide, being more common in CG islands, characteristic of transposons, mainly due to ….

H cannot be every nucleotide. There is a lot of information in this sentence, but references are missing.

Conclusions

line 376-379: “For our new data, new approaches are needed to elucidate the biochemical and molecular mechanisms involved in the methylation of specific during Coffea IS.” … specific… what???

Please complete the sentence.

Author Response

It is with great pleasure that we are resubmitting to Forests the article “Genomic methylated cytosine level during the dedifferentiation and cellular competence in Coffea arabica lines: insights about the different in vitro responses".

All criticisms were considered and the manuscript modified according to Reviewers’ suggestions. According to Forests rules, all changes were highlighted in the revised manuscript, using the fluorescent highlighter “yellow”. Thanks for being so supportive towards our manuscript. Hope to hearing from you soon!

REVIEWER 3

Joao Paulo de Morais Oliveira and Colleagues presented a paper investigating the level of global genomic DNA methylation in Coffea arabica plants. The Authors focused on three aspects: (1) establish the indirect somatic embryogenesis in C. arabica, (b) associate the distinct responses to the global level of DNA methylation, (c) check the stability of ploidy in regenerants.

All of them seem vital and relevant for readers interested in in vitro cultures; however, the authors did not avoid shortcomings. Main note, the manuscript needs linguistic smoothing as some sentences do not fully convey the statement's meaning. However, it seems that improving the language would significantly increase the value of the manuscript.

Abstract

“From these somatic embryos, plantlets were recovered, exhibiting the same 2C value 27 and chromosome number than plant donors”

….than?... There seems to be an inaccuracy in this sentence; please correct it.

Authors’ comments: The manuscript was modified according to all suggestions.

Introduction

“Indirect somatic embryogenesis (ISE) is a morphogenetic pathway that consists of the cultivation of plant tissue segments in sterile and specific culture medium, in the originating callus and, later, in the somatic embryos (SE) and plantlets.”

This sentence is not entirely clear. It needs rewording. ….(ISE)..that consists of… in the originating callus????...

“The somaclonal variation is associated to genomic (nucleus, mitochondria and plastid) and/or epigenomic changes, which can result in phenotypic variations of the regenerated plantlets in relation to the plant explant donors – called genetic fidelity loss (de Oliveira et al. 2019).”

….plant explant donors… or as previously line 53 …explant donor plant…??? It would be good if the Authors adopted one convention to describe donor plants (source of explants) and stuck to this version throughout the manuscript. Therefore, please consider this comment throughout the text.

“These variations….????

Authors write about somaclonal variation, so I guess it should be “This variation…”

Could the Authors specify more precisely what "pedigree" means in this case? Perhaps it would be worth adding 1-2 sentences of explanation.

Authors’ comments: The manuscript was modified according to all suggestions.

I would like to know whether the examples the Authors give from the literature actually refer to somaclonal variation (SV) or perhaps to tissue culture-induced variation (TCIV)? From the definition of somaclonal variation, it is variation induced in vitro and inherited in the generative progeny of regenerants and should be analyzed not in regenerants but in the generative progeny of regenerants. Very often, these terms are used interchangeably, probably not quite rightly.  Hence my question is if the quoted examples concern SV or TCIV?

Authors’ comments: In the revised manuscript, we showed the somaclonal variation concept and the more used analyses to check the genomic and epigenomic stability. Regarding to indirect somatic embryogenesis, some studies evaluated the somaclonal variation occurrence during the in vitro response, including the differences between the callus cells. As several studies, we evaluated the karyotype stability of the regenerated plantlets of two C. arabica lines. For this, flow cytometry has been widely accomplished, but the chromosome counting is also indicated because this data evidence the 2n chromosome number of the cells or plantlets.

Statistical analysis

I am curious why the Authors have such a large discrepancy in repetitions of 8, 9, 20, or 6 and 20?

Authors’ comments: The measurement of the 5-mC requires good quality and sufficient quantity of the genomic DNA. So, the discrepancy between the repetitions is related to the availability of biological material to extract adequate genomic DNA.

Results

The description of statistical analyzes of results lacks whether there were significant outliers in the groups of the independent variable in terms of the dependent variable, whether the dependent variable was approximately normally distributed for each group, and whether the homogeneity of variances was preserved? It would be good to provide the results of the appropriate tests checking the above issues.

Authors’ comments: As observed in the box plot graphs, there were outliers, since intraspecific variations were observed for each lineage. However, the data were transformed by √ (x + 0.5) and the presence of outliers did not affect the homogeneity of the variances. For this reason, the outliers were removed from the figure.

ANOVA Calogenesis (Figure 1).

GL

SQ

QM

Fc

Pr>Fc

Day

5

0.840

0.168

1.97

0.000e+00

Lines

2

94.051

47.025

552.82

0.000e+00

Day*Lines

10

0.150

0.015

0.18

0.00997

Residue

1224

104.120

0.085

Total

1241

199.161

CV = 12.91 %

Waste normality test (Shapiro-Wilk)

p-value:  1.195374e-29

ATTENTION: at 5% significance, waste cannot be considered normal!

ANOVA Embryogenesis (Figure 2).

GL

SQ

QM

Fc

Pr>Fc

Day

5

364.31

72.86

505.3

0.000e+00

Lines

1

865.93

865.93

6005.1 

0.000e+00

Day*Lines

5

300.87

60.17

417.3

6.305e-273

Residue

1380

199.00

0.14

Total

1391

1730.10

CV = 24.42 %

Waste normality test (Shapiro-Wilk)

p-value:  3.895438e-45

ATTENTION: at 5% significance, waste cannot be considered normal!

ANOVA Methylated Cytosine (Figure 4).

GL

SQ

QM

Fc

Pr>Fc

Day

1

419.6 

419.57

4.7036

0.033822

Lines

2

3245.8

1622.89

18.1932

0.000001

Day*Lines

2

924.4

462.19

5.1814

0.008211

Residue

64

5709.0

89.20

Total

69

10298.7

CV = 27.45 %

Waste normality test (Shapiro-Wilk)

p-value:  0.009323955

ATTENTION: at 5% significance, waste cannot be considered normal!

line 207: typo  …'Catuaí Vemelho…

lines 212-214: Is a hyphen before the figure caption needed in all figures? I would move the phrase “The highest mean number of responsive explants was observed in 'Oeiras', followed by 'Caturra' and 'Catuaí Vermelho' for all evaluated times.” to the description of the results rather than leaving it in the figure caption.

line 232:…As in the comment above, in the figure caption, I would leave only the relevant description concerning the graph and pictures and move the descriptions to the results text.

line 238:… “Leaf explants responded to the means of induction and proliferation, resulting in friable callus. However, friable calli did not acquire embryogenic competence and remained in a stage of callus.3.1. Subsection.” I would move this text to the description of the results.

Authors’ comments: The manuscript was modified according to all suggestions.

Discussion

line 331: “Our data show that the in vitro tissue culture condition alters the methylation patterns, …” I am not entirely convinced that the Authors should use the phrase "methylation pattern"; after all, they investigated the global level of methylation in genomic DNA rather than the location of methylated cytosines, e.g., in specific CG, CHG, or CHH sequences. Therefore, I would be careful with such a statement.

line 337: “….from a eucromatic to a…” typo

Lines 346-349: This high degree of cytosine methylation in plants is attributed to the fact that, in plants, methylation can be present in three CG islands, CHG and CHH, where H can be any deoxynucleotide, being more common in CG islands, characteristic of transposons, mainly due to ….

H cannot be every nucleotide. There is a lot of information in this sentence, but references are missing.

Authors’ comments: The manuscript was modified according to all suggestions.

Conclusions

line 376-379: “For our new data, new approaches are needed to elucidate the biochemical and molecular mechanisms involved in the methylation of specific during Coffea IS.” … specific… what???

Please complete the sentence.

Authors’ comments: The sentence was completed.

Reviewer 4 Report

See following comments for this manuscript.

  • More information on cytosine methylation with references is required in Introduction section and justify the title.
  • All scientific names should be in italic.
  • Add media information and other details in the ISE establishment section.
  • What id friable callus? Is it embryogenic callus? make it clear in method section as well as through images. 
  • Make a separate section for flowcytometry and and microscopy.
  • What is difference between figure 1 A and B? I think it is confusing, better to remove figure 1B and increase the size for the calli pictures. Same as figure 2 B.
  • Figure 2. Images for the embryoids stages are not clear. Make it bigger or put them in the separate figure.
  • Figure 3. Same comment as above. 
  • Elaborate the discussion section.

Author Response

It is with great pleasure that we are resubmitting to Forests the article “Genomic methylated cytosine level during the dedifferentiation and cellular competence in Coffea arabica lines: insights about the different in vitro responses".

All criticisms were considered and the manuscript modified according to Reviewers’ suggestions. According to Forests rules, all changes were highlighted in the revised manuscript, using the fluorescent highlighter “yellow”. Thanks for being so supportive towards our manuscript. Hope to hearing from you soon!

REVIEWER 4

More information on cytosine methylation with references is required in Introduction section and justify the title.

All scientific names should be in italic.

Authors’ comments: The manuscript was modified according to all suggestions.

Add media information and other details in the ISE establishment section.

Authors’ comments: Additional information requested is described in the supplement.

What id friable callus? Is it embryogenic callus? make it clear in method section as well as through images.

Authors’ comments: The manuscript was modified as per suggestion.

Make a separate section for flowcytometry and and microscopy.

Authors’ comments: There is no need, as these analyzes are complementary.

What is difference between figure 1 A and B? I think it is confusing, better to remove figure 1B and increase the size for the calli pictures. Same as figure 2 B.

Authors’ comments: Figure 1A as well as Figure 2A show the result of the mean test. While Figures 1B and 2B show the result of the regression analysis.

Figure 2. Images for the embryoids stages are not clear. Make it bigger or put them in the separate figure.

Figure 3. Same comment as above.

Authors’ comments: The resolution of the figures has been improved.

Elaborate the discussion section.

Authors’ comments: The discussion has been changed and elaborated.

Reviewer 5 Report

Although the topic is attractive, there are some concerns that should be addressed. The manuscript lacks very important points which should be included for the plant researchers and tissue culture scientists.

L 3: "Coffea arabica" should be italics.

L 13: "Coffea arabica" should be italics.

PLEASE WRITE ALL THE PLANT BOTANICAL NAMES IN ITALICS.

L 14-15: For other species? How about the other crops? DNA methylation can be considered as one of the most important epigenetic machinery in all plants during in vitro culture. Please revise the sentence.

L 50-52: Please provide reference(s): (https://doi.org/10.1002/reg2.91; https://doi.org/10.1007/s00253-021-11375-y; https://doi.org/10.3390/ijms21072307)

L63-65: Please provide more new references: (https://doi.org/10.1146/annurev-arplant-050718-100434; https://doi.org/10.3390/ijms22115671; https://doi.org/10.1016/j.tplants.2017.11.009)

Material and methods should be presented in more detail. For example, please provide the brand of the applied chemicals (e.g., media, PGRs, etc.).

Discussion must be improved. Please discuss the role of PGRs and different types of callus (compact, friable, etc.) on somatic embryogenesis.

Discussion must be improved. Please discuss the role of PGRs and different types of callus (compact, friable, etc.) on somatic embryogenesis. (https://doi.org/10.1016/j.eng.2018.11.006; https://doi.org/10.1007/s00253-021-11375-y)

Discussion must be improved. Please discuss the role of PGRs and different types of callus (compact, friable, etc.) on somatic embryogenesis. (https://doi.org/10.1016/j.eng.2018.11.006; https://doi.org/10.1007/s00253-021-11375-y)

The conclusion section is very short. At least it should discuss more future work.

Author Response

It is with great pleasure that we are resubmitting to Forests the article “Genomic methylated cytosine level during the dedifferentiation and cellular competence in Coffea arabica lines: insights about the different in vitro responses".

All criticisms were considered and the manuscript modified according to Reviewers’ suggestions. According to Forests rules, all changes were highlighted in the revised manuscript, using the fluorescent highlighter “yellow”. Thanks for being so supportive towards our manuscript. Hope to hearing from you soon!

REVIEWER 4

Although the topic is attractive, there are some concerns that should be addressed. The manuscript lacks very important points which should be included for the plant researchers and tissue culture scientists.

L 3: "Coffea arabica" should be italics.

L 13: "Coffea arabica" should be italics.

PLEASE WRITE ALL THE PLANT BOTANICAL NAMES IN ITALICS.

Authors’ comments: All botanical plant names have been placed in italics.

L 14-15: For other species? How about the other crops? DNA methylation can be considered as one of the most important epigenetic machinery in all plants during in vitro culture. Please revise the sentence.

L 50-52: Please provide reference(s): (https://doi.org/10.1002/reg2.91; https://doi.org/10.1007/s00253-021-11375-y; https://doi.org/10.3390/ijms21072307)

L63-65: Please provide more new references: (https://doi.org/10.1146/annurev-arplant-050718-100434; https://doi.org/10.3390/ijms22115671; https://doi.org/10.1016/j.tplants.2017.11.009)

Authors’ comments: The manuscript was modified according to all suggestions.

Material and methods should be presented in more detail. For example, please provide the brand of the applied chemicals (e.g., media, PGRs, etc.).

Authors’ comments: This request has already been provided in the supplement and throughout the methodology.

Discussion must be improved. Please discuss the role of PGRs and different types of callus (compact, friable, etc.) on somatic embryogenesis  (https://doi.org/10.1016/j.eng.2018.11.006; https://doi.org/10.1007/s00253-021-11375-y).

Authors’ comments: The manuscript was modified according to all suggestions.

The conclusion section is very short. At least it should discuss more future work.

Authors’ comments: The conclusion was modified and we showed some perspectives for further studies.

Round 2

Reviewer 2 Report

In my opinion, article is not improve after review. The manuscripts not include any fluorescent highlighter “yellow”.  The change level of m5C in responsive and non-responsive calli not explain any mechanism function of 5mC in genes expression during dedifferentiation and somatic embryogenesis. Authors didn’t add additional results concern of somaclonal variation was investigated for all cultivars even to supplementary data.

Reviewer 4 Report

In method section, add information on cultivars (Caturra', 'Oeiras' and 'Catuaí Vermelho) used in the study for their characteristics.

Provide detail information in method section like total numbers of leaf explants per cultivar and method for counting the number of responsive explants. Was it based on Petri-dish or cultivar? Same for the MCSE

What is the ploidy level of mother (donor) plants 'Oeiras'. 

In figure 5, Specify the cultivar's name for the count and chromosome number.

Line 374-75, justify the statement reproducible. How many times the experiments repeated?

Reviewer 5 Report

All my comments have been addressed. I think that this version of the MS can be published in Forests.